# Conservation Genomic Analysis of the Asian Honeybee in China Reveals Climate Factors Underlying Its Population Decline

**DOI:** 10.3390/insects13100953

**Published:** 2022-10-19

**Authors:** Huiling Sang, Yancan Li, Cheng Sun

**Affiliations:** 1Institute of Apicultural Research, Chinese Academy of Agricultural Sciences, Beijing 100093, China; 2College of Life Sciences, Capital Normal University, Beijing 100048, China

**Keywords:** Asian honeybee, *Apis cerana*, conservation genomics, effective population size

## Abstract

**Simple Summary:**

The Asian honeybee is an important pollinator in Asia that plays a vital role in maintaining biodiversity. Based on field surveys and personal observations, Asian honeybee populations in China were reported to be undergoing significant decline in 2005. However, a more recent survey revealed that its populations are stable and even slightly increased in some regions of China. Therefore, the status of declining Asian honeybee populations in China is still unclear. In this study, taking advantage of the abundant genomic data for Asian honeybees, we employed conservation genomics methods to understand the declining status of Asian honeybee populations in China and identify the causing factors. We found that most of the Asian honeybee populations of China showed a relatively stable population size during recent years, however, the population in Bomi, Tibet was in a serious decline and low temperatures and strong ultraviolet radiation should have synergistically led to this decline. Our study provides insights into the dynamic changes of Asian honeybee populations in China and identifies climate factors that underlie its population decline, which are valuable for the conservation of this important pollinator.

**Abstract:**

The Asian honeybee, *Apis cerana*, is one of the most important native pollinators in Asia. Asian honeybees were believed to be under significant decline in China based on a report in 2005. On the contrary, a recent survey revealed that Asian honeybee populations in China are stable and even slightly increased in some regions. Therefore, the declining status of *A. cerana* populations in China is still unclear. Taking advantage of the abundant, publicly available genomic data for Asian honeybees in China, we employed conservation genomics methods to understand if Asian honeybee populations in China are declining and what the underlying climate factors are. We reconstructed the changes of effective population size (*Ne*) within the recent past for 6 population groups of Asian honeybees and found out that only one of them (population in Bomi, Tibet) showed a consistently declining *Ne* from the last 100 generations to 25 generations. Selective sweep analysis suggests that genes related to the tolerance of low temperatures and strong ultraviolet radiation are under selection in the declining population, indicating that these two climate factors most likely underlie the decline of BM populations during the recent past. Our study provides insights into the dynamic changes of Asian honeybee populations in China and identifies climate factors that underlie its population decline, which is valuable for the conservation of this important pollinator.

## 1. Introduction

The Asian honeybee (*Apis cerana*) is one of the most important native pollinators of natural and agricultural cropping systems in Asia [1]. *Apis cerana* is capable of finding and visiting sporadic nectar sources and foraging, even when the ambient temperature is low, which indicates *Apis cerana* plays a pivotal role in food production, human livelihoods and biodiversity [2,3]. Based on a report in 2005, *A. cerana* populations in China have been in significant decline: its distribution area has shrunk by more than 75% and the population size has decreased by more than 80% [2]. The decline of bee populations will destabilize the ecosystem, bringing great losses to agricultural production and causing food shortages [4,5]. There is no doubt that the population decline of honeybees has aroused widespread concern. However, a recent survey revealed that Asian honeybee populations in China are stable and even slightly increased in some regions [6]. Therefore, the status of declining *A. cerana* populations in China during the recent past is still unclear.

In the past, researchers mainly relied on field surveys and historical records taken at two different periods to understand whether a specific population size in a given habitat has declined or not [2,6]. However, this traditional approach has many drawbacks. Firstly, the process is time-consuming and labor-intensive. Secondly, the results obtained from the two different periods may be incomparable due to human and technical differences. In addition, although a population decline could be found in this way, the exact causes of this decline cannot be determined.

Conservation genomics is an emerging field that applies whole-genome resequencing data to the preservation of the viability of populations and the biodiversity of living organisms [7,8]. Conservation genomics provides new methods for excavating the information contained in the whole genome of a species or population to estimate the historical dynamic changes of the effective population size (*Ne*), determine whether the population declined and identify the causes underlying the population decline [9,10,11,12]. Effective population size (*Ne*) is a key genetic parameter that estimates the amount of genetic drift in a population, and has been described as the number of individuals of an ideal population exhibiting the same level of genetic drift as the studied population [13,14]. Conservation genomics has been widely used in species conservation due to its many advantages. For example, by using conservation genomics, researchers can identify genes that may have played a role in adapting to environmental stressors underlying the bumblebee *Bombus terricola*’s decline in North America [15].

In this study, taking advantage of the abundant, publicly available genomic sequences for 155 Asian honeybees, which were distributed in China and can be divided into six population groups, we employed conservation genomics methods to understand the status of decline of Asian honeybee populations during the recent past and identify the causal factors. Our results reveal that, while the overall Asian honeybee populations in China are stable, low temperatures and strong ultraviolet radiation are detrimental to the population size of this important pollinator.

## 2. Materials and Methods

### 2.1. Whole Genome Shotgun Reads

The genome resequencing data for 155 Asian honeybee (*Apis cerana*) workers were downloaded from the National Center for Biotechnology Information (NCBI) Sequence Read Archive (SRA) database (https://www.ncbi.nlm.nih.gov/sra, accessed on 20 November 2021), which were mainly generated by two previous studies [16,17]. The selected 155 workers belong to six different population groups of Asian honeybee according to a previous report [16], namely Bomi (BM; Bomi, Tibet), Aba (AB; Maerkang, Aba prefecture, Sichuan province), Qinghai (QH; Xunhua and Guide, Qinghai province), Hainan (HN; Hainan Island), Northeast China (NE; including Heilongjiang, Jilin and Liaoning province) and Shennongjia (SNJ; Shennongjia, Hubei province). We selected these six Asian honeybee population groups because their geographical distribution is wide enough to have abundant variation in climate factors across different groups and the number of sequenced honeybee individuals in each group was enough to accurately reconstruct their recent change trends of effective population size (see Methods). The numbers of sequenced Asian honeybees for each group are summarized in Table 1, with detailed information available in Appendix A. In addition, the reference genome sequence for the Asian honeybee of the central group, the ancestral group [16], was downloaded from NCBI SRA database PRJNA806528.

### 2.2. Read Mapping, Variant Calling and Filtering

First, we verified the quality of the raw sequencing reads for 155 honeybee samples using FastQC 0.11.9 (https://www.bioinformatics.babraham.ac.uk/projects/fastqc/, accessed on 14 August 2022), followed by the data quality control using fastp 0.23.2 [18] with the following parameters: qualified quality Phred: 10; unqualified percent limit: 50; and base limit: 10. Next, we mapped the filtered shotgun reads to the reference genome of the Asian honeybee using BWA-MEM 0.7.17 [19], with parameters -t 10 -M. The resulting bam files were sorted and observed duplicates were removed after mapping using Picard 2.26.9 (https://broadinstitute.github.io/picard/, accessed on 14 August 2022). We then used GATK 4.1.8.1 [20] to call variants in the subsequent steps. Specifically, HaplotypeCaller was first used to generate an intermediate file called GVCF for each sample independently. We then combined all of the obtained GVCFs into one file and passed it to the joint genotyping tool, GenotypeGVCFs, to generate a set of joint-call cohorts (SNP and indel calls). Finally, the variants were first filtered following GATK’s recommended threshold (i.e., QD < 2.0, FS > 60.0, MQ < 40.0), and then were further filtered using VCFtools [21] with the following parameters: maf 0.01, max-missing 0.5 -min–meanDP 2, max–meanDP 40, hwe 0.001, minQ 20, minGQ 20, min-alleles 2, max-alleles 2 and remove-indels.

### 2.3. Estimation of the Recent Effective Population Size

Inferring changes in effective population size (*Ne*) in the recent past is very important for the conservation of focal species, through which we can determine if the species is undergoing population decline. The program GONE can efficiently infer the demographic history of a population within the past 100 generations based on linkage disequilibrium of pairs of loci [22]. In addition, a recent study found that estimates of effective population size based on linkage disequilibrium, the method that GONE performs, are virtually unaffected by natural selection [23]. In this study, the final filtered VCF file obtained from the previously mentioned procedures was processed by PLINK 1.9 [24] to produce PED and MAP files. The two files then served as input files for GONE to reconstruct changes in *Ne* in the recent past for each population group of the Asian honeybee (*A. cerana*). For this study, the duration of a generation was assumed to be one year. The default parameter settings were applied when running GONE, excepting some cases, including: (a) applying a recombination rate of 17.4 cM/Mb for *A. cerana* [25]; (b) the number of SNPs per chromosome to be analyzed being set to 25k SNPs because of a relative low SNP density of the final filtered SNP dataset; and (c) the highest recombination frequency (*hc*) being set to 0.01 in view of *A. cerana* exhibiting a high migration [17].

For cross-validation purposes, a parametric approximate Bayesian computation approach, PopSizeABC [26], was also used in this study to infer the dynamics of effective population size (*Ne*) in the recent past. The computation of the PopSizeABC approach was based on the allele frequency spectrum (AFS) and the average linkage disequilibrium (LD) at different bins of physical distance between SNP data [26]. For *A. cerana* populations, the parameters were set as follows: the per generation per bp recombination rate was set to 17.4 × 10^−8^, with a range of [17.0 × 10^−8^, 17.4 × 10^−8^] [25]; the mutation rate was set to 5.27 × 10^−9^ per generation per bp [27]; the number of independent segments in each dataset was set to 16; the haploid sample size was set to 2 × (number of individuals) for each population; the number of simulated datasets was set to 500; and all the other parameters were set to default.

### 2.4. Selective Sweep Analysis

In order to detect genomic regions under selective sweep in Asian honeybee populations that reportedly had a declining *Ne* in the recent past, two complementary methods were used. For Method 1, the six population groups were first classified as “declining” or “non-declining.” Next, the “declining” and “non-declining” groups were merged as declining population and non-declining population, respectively. We then calculated the nucleotide diversity (θπ) for both the declining population and the non-declining population, and the pairwise fixation index (*Fst*) between them, with 10-Kb sliding windows and a 5-Kb step size, using VCFtools [21]. Finally, we used BEDTOOLS [28] to extract genomic regions that fell within the intersection of the top 5% of both Z-transformed *Fst* and log value of θπ ratio (ratio of non-declining population to declining population). Genes lying within those genomic regions were deemed as under selection in the honeybee populations that had a declining *Ne* in the recent past. For Method 2, we first identified honeybee groups that had a declining *Ne* and merged them as one declining population. We then calculated the θπ and *Fst* values between the declining population and each of the other non-declining populations and identified genomic regions that fell within the intersection of the top 5% of Z-transformed *Fst* and log value of θπ ratio (ratio of non-declining population to declining population). Genes lying within identified genomic regions were extracted for each pair of comparison, and only overlapped genes across all comparisons were deemed as under selection in Asian honeybee populations that had a declining *Ne*. Identified genes under selection from each of the two methods were conducted in gene-list enrichment analysis using the enrichment module of KOBAS-i [29]. Specifically, we selected *Apis mellifera* (honeybee) from the species list provided by the tool’s website as a background list, and the input gene list was the protein sequence of genes under selection (in FASTA format).

### 2.5. Extraction of Bioclimatic Variables

To understand the effect of climatic factors on the recent population dynamics of *A. cerana*, we first downloaded the “.tif” files for average temperature and solar radiation (between 1970 and 2000) from WorldClim 2.1 database (www.worldclim.org) at a resolution of 30 seconds, and then extracted average temperature and solar radiation data for each population group using ArcMap10.2 (www.esri.com).

## 3. Results and Discussion

### 3.1. Population Decline Was Observed in One of the Six Asian Honeybee Population Groups

The genomic shotgun reads of 155 honeybee samples were mapped to the reference genome sequence of the central group (ancestral group) of Asian honeybees [16], resulting in an average mapping depth of approximately 15× (Appendix A). After strict filtering, the final variant-calling dataset consisted of 550,218 single nucleotide polymorphisms (SNPs). As the shotgun reads used in this study were derived from two previous studies [16,17], before estimating the recent effective population size, we performed principal component analysis (PCA) to determine if there was any batch effect. The PCA result showed that honeybee samples were clustered by population group (Figure 1), but not by previous research, therefore no batch effect was detected between the two studies.

We then utilized the software GONE to infer the *Ne* change within the last 100 years for each of the six population groups. Because extended sampling generation or species migration could lead to a bias of *Ne* estimation for the first quarter of the sampling period [22], we disregarded *Ne* estimations from the time of sampling to 25 generations. From the results, we could see that, although there are fluctuations, five out of the six population groups (AB, QH, HN, NE and SNJ) exhibit an increasing or relatively stable *Ne* from 25 to 100 generations ago (equivalent to 25–100 years ago, as *A. cerana* is 1 year/generation), and only the BM population showed a consistently declining *Ne* in the recent past (Figure 2). For cross-validation and comparison purposes, another software, PopSizeABC, was also used in this study to infer recent changes in *Ne*. The results of PopSizeABC for each population group were almost consistent with that of GONE within the last 25–100 generations, except within the QH population group (Figure 3). Taking into consideration the accuracy when inferring *Ne* in recent timeframes, GONE outperforms other leading approaches, including PopSizeABC [22], thus we classified the QH population as a non-declining group during the recent past.

In addition, the two software that we employed in this study, GONE and PopsizeABC, are complementary to each other. While GONE has an advantage in estimating recent *Ne*, PopsizeABC does well in monitoring a focal species’ demographic history in the long-term [22,30]. Reviewing the results of PopsizeABC (Figure 3), we could see that AB, HN, NE and SNJ population groups showed a relatively stable *Ne* during a long period of time (although there were population declines, a recovery occurred after each decline). Coupled with results from GONE (Figure 2), which showed a stable or even increased *Ne* for these four population groups, we could conclude that the four population groups of Asian honeybees in China (AB, HN, NE and SNJ) are of little concern for declining population. Regarding the QH population, while PopSizeABC indicated a population decline began at least 1000 years ago (Figure 3), the result from GONE showed an increasing *Ne* within the last 100 generations (Figure 2), which may suggest that, although it has experienced a long-term population decline, QH population has begun to recover in the recent past. Regarding the BM population, both PopSizeABC and GONE analysis revealed a drastic population decline beginning thousands of years ago to the last 100 generations (Figure 2 and Figure 3), indicating that the BM population requires a great deal of attention for conservation purposes.

Previous reports on the changes of Asian honeybee populations were mainly based on personal observations and experiences [2,6], whereas our study represents the first application of conservation genomics methods to understand the population dynamics of this important pollinator. Our results indicate that during the past several decades, the overall Asian honeybee populations in China were stable or in a recovery phase (Figure 2 and Figure 3), which is mostly consistent with the results obtained from a recent survey of Asian honeybee in 16 countries [6]. However, the BM population of Asian honeybee in China is in a serious decline, and people need to take immediate conservation measures.

### 3.2. Genes under Selection in a Population with Declining Ne

We used two methods to search for genes under selective sweep in the BM population based on θπ and *Fst* values (see Methods). For Method 1, when we compared the BM population, the only honeybee group with declining *Ne*, with the combined population of the other five non-declining population groups, a total of 50 genes were found under selection in the BM population (Figure 4). After performing KEGG enrichment analysis on these genes, two enriched KEGG pathways (*p* value <= 0.05), alpha-linolenic acid metabolism and one carbon pool by folate, were identified (Table 2, Figure 5). The alpha-linolenic acid (alphaLNA; 18:3n-3) is a precursor of the synthesis of polyunsaturated fatty acids (PUFAs), which are long chain fatty acids that play an important role in maintaining proper biological membrane fluidity [31,32]. Organisms demand more PUFAs when exposed to low temperatures, thus leading to an increased synthesis of alpha-linolenic acids [33]. In other words, the alpha-linolenic acid metabolism pathway is related to cold tolerance. Another enriched KEGG pathway, one-carbon (1C) metabolism mediated by the folate cofactor, plays a role in the repair of DNA damage caused by exposure to high ultraviolet radiation (UVR) [34,35,36]. Therefore, genes in alpha-linolenic acid and one carbon pool by folate metabolism pathways were under selection in the only declining population, indicating that the BM population has been subjected to low temperatures and strong ultraviolet radiation in its habitat, resulting in a declining *Ne* in the recent past.

For Method 2, when comparing the declining BM group with each of the other five non-declining groups to identify genes under selection, a total of 8 genes were detected and then enriched in four pathways (*p* value < 0.05): mismatch repair, DNA replication, nucleotide excision repair and inositol phosphate metabolism (Table 3). Mismatch repair, DNA replication and nucleotide excision repair pathways are all related to DNA repair mechanism triggered by DNA damage that occurs as a result of exposure to exogenous chemicals and physical agents, such as polychlorinated biphenyls and strong ultraviolet light [37]. The inositol phosphates usually act as second messengers for a variety of extracellular signals [38]. Among them, myo-inositol (MI) is known to be one of the main cryoprotectants in insects to help them withstand low temperatures [39,40,41]. Therefore, these four enriched pathways indicate that the decline of BM population could be caused by the combined effects of ultraviolet radiation and low temperatures, and these results are consistent with that of Method 1.

The results of our two complementary methods are consistent, indicating that low temperatures and strong ultraviolet radiation might be the underlying reasons for the decline of the BM population.

### 3.3. Low Temperatures and High Ultraviolet Radiation May Synergistically Affect Asian Honeybees

Temperature is one of the most important factors in insect development, and suboptimal temperatures have been shown to affect the survival and development of bee broods [42]. During cold months, Asian honeybees form a cluster, clinging tightly together on the combs in the hive [43] to protect themselves from heat loss [44,45], while in summer, they are most active, engaging in a variety of activities. Therefore, the ambient temperature in summer has a more direct influence on Asian honeybees. High levels of ultraviolet radiation have been shown to affect the development and survival of insects [46], which could affect Asian honeybees all year round.

To further understand the influence of low temperatures and strong ultraviolet radiation on the BM population, we extracted and summarized the mean summer temperature (June, July and August) and the average year-round solar radiation for geographic regions where the six honeybee population groups reside. We found that, compared to the other five population groups, the BM population has the second lowest mean temperature in summer and the second highest solar radiation throughout the year (Table 1; detailed information in Appendix A), and no other population group has a similar combination. Therefore, it is likely that a synergistic effect of low temperature and high ultraviolet radiation has caused the decline of the BM population, which is consistent with the result of our genome-wide selective sweep analysis.

Conservation measures, such as providing shelter and shade, could be taken when performing colony management to prevent Asian honeybees in Bomi, Tibet from suffering from low temperatures and strong ultraviolet radiation, which could protect them from further population decline.

## 4. Conclusions

In this study, taking advantage of the abundant, publicly available genomic sequences for Asian honeybees, we employed conservation genomics methods to understand the declining status of Asian honeybee populations in China and identify the causal factors. Our results revealed that the population sizes for five out of the six Asian honeybee population groups in China were relatively stable, and only one Asian honeybee population group (Bomi, Tibet) showed a consistently declining population size, for which people should take immediate conservation measures. We found out that genes related to the tolerance of low temperatures and ultraviolet radiation are under selection in the declining population, indicating that these two climate factors could synergistically lead to the population decline of the Asian honeybee.

## Figures and Tables

**Figure 1 insects-13-00953-f001:**
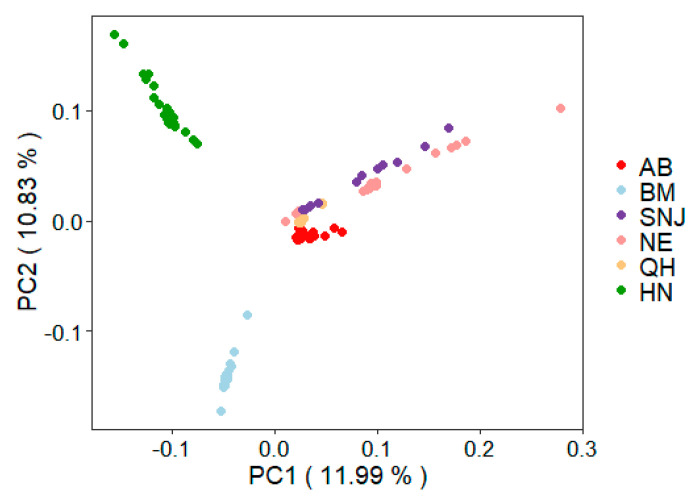
PCA plots of the 155 honeybee samples (the first two PCs). Honeybee samples were clustered by population groups (AB from Aba prefecture, Sichuan province; BM from Bomi, Tibet; SNJ from Shennongjia, Hubei province; NE from Liaoning, Jilin and Heilongjiang province; QH from Qinghai province; HN form Hainan Island).

**Figure 2 insects-13-00953-f002:**
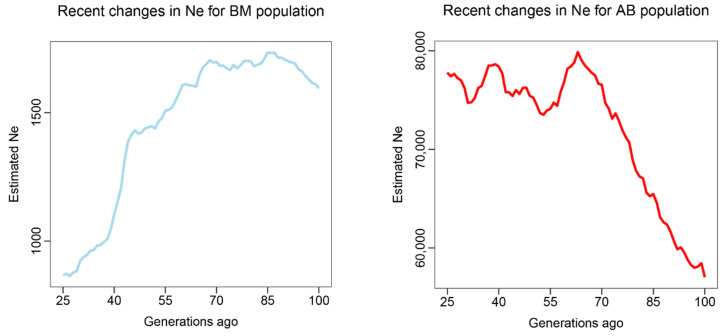
The trends of effective population size (*Ne*) change in the recent past inferred by software GONE for six Asian honeybee population groups (BM from Bomi, Tibet; AB from Aba prefecture, Sichuan province; QH from Qinghai province; HN form Hainan Island; NE from Liaoning, Jilin and Heilongjiang province; SNJ from Shennongjia, Hubei province).

**Figure 3 insects-13-00953-f003:**
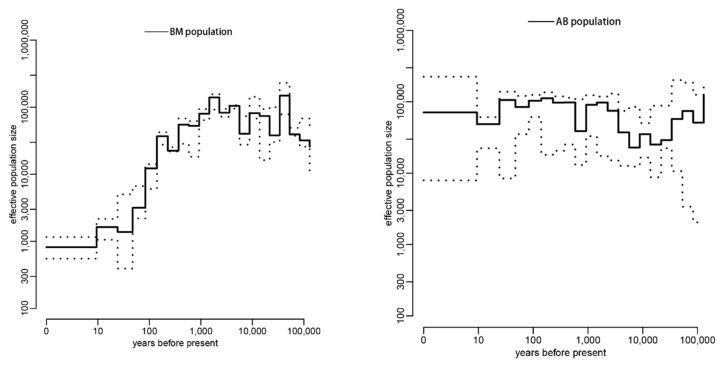
The recent effective population size (*Ne*) for each population group inferred by software PopSizeABC. A 95% credible interval is indicated by dotted lines.

**Figure 4 insects-13-00953-f004:**
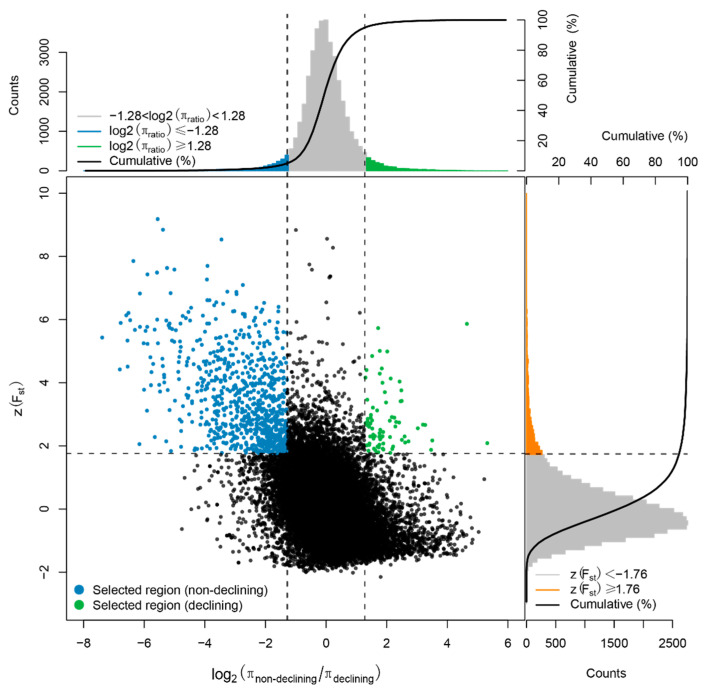
The distribution of candidate genomic regions detected by selective sweep analysis between non-declining populations and the declining population (BM population) based on θπ and *Fst*. Green points were identified as selective sweeps in the declining population (BM population) that passed the thresholds of log θπ ratio (the 5% right tail of the empirical θπ ratio distribution, log2(π ratio) ≥ 1.28) and Z-transformed *Fst* (the 5% right tail of the empirical *Fst* distribution, Z(*Fst*) ≥ 1.76).

**Figure 5 insects-13-00953-f005:**
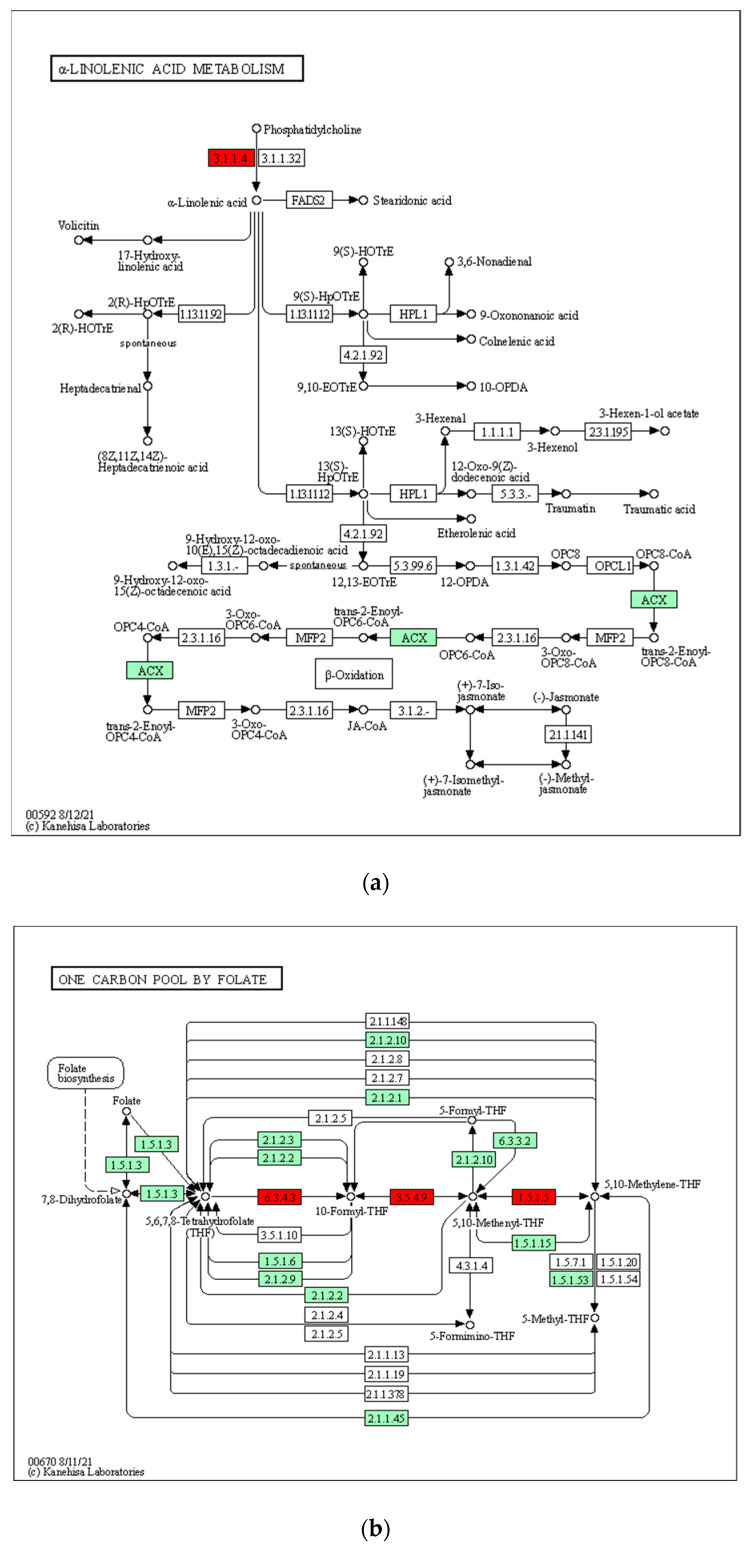
Two enriched KEGG pathways for genes under selection detected in Method 1. Gene numbers with a red background represent genes under selection. Figure (**a**) represents the alpha-linolenic acid metabolism pathway; figure (**b**) represents the one carbon pool by folate pathway.

**Table 1 insects-13-00953-t001:** Number of sequenced Asian honeybees for each population group.

Population Group	Number of Individuals	Mean Elevation (m)	Mean Temperature for Summer (°C)	Mean Solar Radiation Year-Round (kJ m^−2^ day^−1^)
BM	29	2360.42	17.02	15,253.02
AB	28	2652.83	14.55	12,970.50
QH	20	2154.5	17.18	15,178.33
NE	29	369.32	21.21	14,839.49
HN	34	133.58	28.33	17,014.26
SNJ	15	1637.6	18.83	13,238.05

**Table 2 insects-13-00953-t002:** KEGG enrichment analysis of genes under selection of Method 1.

Term	Database	ID	*p*-Value
alpha-linolenic acid metabolism	KEGG PATHWAY	ame00592	0.039084
One carbon pool by folate	KEGG PATHWAY	ame00670	0.046028
Arachidonic acid metabolism	KEGG PATHWAY	ame00590	0.063174
Ether lipid metabolism	KEGG PATHWAY	ame00565	0.066567
Mismatch repair	KEGG PATHWAY	ame03430	0.069948
Protein export	KEGG PATHWAY	ame03060	0.073317
Basal transcription factors	KEGG PATHWAY	ame03022	0.116042
DNA replication	KEGG PATHWAY	ame03030	0.122442
Nucleotide excision repair	KEGG PATHWAY	ame03420	0.128797
Toll and Imd signaling pathway	KEGG PATHWAY	ame04624	0.131957
Inositol phosphate metabolism	KEGG PATHWAY	ame00562	0.159902
Apoptosis-fly	KEGG PATHWAY	ame04214	0.162952
Glycerophospholipid metabolism	KEGG PATHWAY	ame00564	0.18992
RNA degradation	KEGG PATHWAY	ame03018	0.192864
Peroxisome	KEGG PATHWAY	ame04146	0.195797
Purine metabolism	KEGG PATHWAY	ame00230	0.238563
Endocytosis	KEGG PATHWAY	ame04144	0.332338
Metabolic pathways	KEGG PATHWAY	ame01100	0.340409

**Table 3 insects-13-00953-t003:** KEGG enrichment analysis of genes under selection of Method 2.

Term	Database	ID	*p*-Value
Mismatch repair	KEGG PATHWAY	ame03430	0.0131
DNA replication	KEGG PATHWAY	ame03030	0.0235
Nucleotide excision repair	KEGG PATHWAY	ame03420	0.0248
Inositol phosphate metabolism	KEGG PATHWAY	ame00562	0.0312
Endocytosis	KEGG PATHWAY	ame04144	0.0709
Metabolic pathways	KEGG PATHWAY	ame01100	0.426

## Data Availability

The raw sequence data analyzed in this study can be found in the NCBI SRA database under accession numbers PRJNA592293, PRJNA418874 and PRJNA869845. The vcf file we used in this study was made publicly available in figshare (https://figshare.com/articles/dataset/Apis_cerana_SNP_clean_datas/21268662/1, accessed on 14 August 2022).

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
