# Peer review of "Conservation Genomic Analysis of the Asian Honeybee in China Reveals Climate Factors Underlying Its Population Decline"

_insects, 2022, doi:10.3390/insects13100953_

Round 1

Reviewer 1 Report

The work aims to elucidate, albeit in part, the dynamic changes of Asian honeybee populations in China and identified climate factors underlying the population decline. For this, the authors used the genome resequencing data of 155 Asian honeybees. For the use of the data, specifically, a correct data processing is needed, which would be my main concern. I made specific comments throughout the manuscript.

Comments

The data used in this study were mainly generated by two researches which is bound to produce different batch effects, can the two batches of data be analyzed directly together? How do you process data to avoid batch effects. Please explain it.

The authors carry out KEGG analysis but provide no information as to how this was done, nor what the background list used was. This is crucial information, if the wrong background list is used (for example wall the genes in the honeybee genome) then this data will be erroneous.

There is no data availability statement in the manuscript. There must be lots of mapping reads and SNP information generated during data processing. These data should be made available. You can show this data in the attachment.

Reference 20-21 have done studies related to rapid radiation and independent adaptation to diverse habitats in the Asian. What is the difference between what they have done and the study you did. You need to add some sentences to clarify the differences. What are the shortages of other studies and what are the strength of yours?

Full textThere are some grammar mistakes, suggest English editing.

Line 38-54: The opening section of the Introduction is very repetitive.

Line 21 Asian 20 honeybees in China were believed to be under significant decline based on a report in 2005. Relevant literature should be inserted here.

Line 109: Please specify what version of the reference genome. Hopefully it was the more recent assembly.

Line 122: The Inferring changes in effective population size (Ne) in the recent past is very important for the conservation of focal species. Please explain this view.

Line 181: How do you judge populations were stable or declining?

Line 207: Please comment the figure 2 in detail. What does the different color mean in this figure? The same problem existed in figure 3.

Line 218: There are two “.” in this sentence.

Reviewer 2 Report

The authors present an interesting study for Asian honeybee populations in China. The authors used publicly available genomic sequences for 155 Asian honeybees, which were distributed in China and could be divided into six population groups. Their results confirm that honeybee populations in China are stable.

The part Results and Discussion is well organized. The Figures and Tables are clear. In my opinion the English is clear. I don't feel qualified to judge about the grammatical and very specific English language and style mistakes. I did not find any technical errors.

Reviewer 3 Report

The authors conducted analyses on the effective population size of six Asian honeybee populations from China. In contrast to previous observations that there may be an overall decline of the Asian honeybee population in China, the author found that only one out of six populations declined in the last 25-100 years. The declining population BM is the one from Bomi, Tibet, which an unique place where many mammals and birds showed adaptive evolution compared the their relatives in other parts of China. The authors then compared the BM population with other populations (both combined and individually) and found that genes or pathways related to DNA repair and inositol phosphate metabolism were under selection. The authors conclude that the decline of the BM population should be caused by the combined effects of ultraviolet radiation and low temperature. The findings in this study is very interesting and important for the conservation of the Asian honeybee population in China. I have two concerns that need to be addressed before publishing the manuscript in insects.

1, The estimation of effective population size is sensitive to the accuracy of molecular markers and methods that used inference the population size. In this study, the authors report only the results of one tool. It would be much better if the authors could confirm the results with other tools or methods.

2, The authors conclude the decline of the BM population should be caused by the combined effects of ultraviolet radiation and low temperature. These two factors are also selection signatures that were identified in mammals and birds. My point is these factors may not be necessary the cause for the decline of the Asian honeybee population in BM population. They may represent selection signatures that Asian honeybees adapted to high altitudes in Tibet. There must be other factors that are associated with the declining population of BM.

Reviewer 4 Report

1. title, change Asian honeybees to Asian honeybee.

2. Abstract: since the status in the past 25 years could not be inferred, could you say the BM population declining consistently?

3. Table 1: provide the GPS, elevation, and mean summer temperature of each site.

4. Figure 3: the font size is too small. the left and right panels are not equal in area and looks unsightly.

5. Conclusion: Are the combined effects of UV and low temperature stronger in the recent past? Why the adaptability could not work on honeybees recently? Does similar phenomenon find in other insect or vertebrate species, or other places of Tibetan Plateau? Is it the consequences of global warming, pollution or other elusive reason?

Round 2

Reviewer 1 Report

I don't have further questions for this paper.